# The effects of icon design features on user perception and preference: A case study of icons for Covid-19

**Deng Lujie**[1], **Chunhua Lin**[2]*, **Qiong Liao**[3], **Shuicai Qiu**[4]

**1** School of the Arts, Universiti Sains Malaysia, Penang, Malaysia, **2** School of Art/Pearl River Film Academy, Jinan University, Guangzhou, China, **3** School of Art and Design, Guangzhou Panyu Polytechnic, Guangzhou, China, **4** School of Arts and Culture, Guangdong Vocational Institute Of Public Administration, Guangzhou, China

* linsarah66@gmail.com

**Data Availability Statement:** All relevant data are within the paper and its Supporting Information files.

**Funding:** The author(s) received no specific funding for this work.

## Abstract

The objective of this study is to evaluate users' perceptions and preferences on the design features of the COVID-19 prevention promotion icon from the perspective of users' aesthetic and perceptual needs. In this study, 120 officially published icons from 24 countries and regions were collected from online platforms for ranking tests, and then the top-ranked icons were subjectively rated by the semantic differential method. By evaluating the quality of users' perceptions of multiple semantic dimensions of icons, we extracted the perceptual semantic words that users valued as the main icon design features. Spearmen correlations were applied to derive possible correlations between user rankings and semantic scales, and a Friedman test was also conducted to determine the true differences in user perceptions and preferences for different styles of icons. Factor analysis was conducted to extract six perceptual words that influence the design features of the COVID-19 prevention promotion icon. The methodology adopted in this study facilitated the screening of design features related to icon effectiveness, and the findings show that "Interesting," "Simple," "Familiar, "Recognizable," "Concrete," and "Close(semantic distance)" are the key features that influence users' perception and preference of COVID-19 icon design. The results of this study can be used as the basis for designing and improving publicity icons for preventive measures in COVID-19, and the methods adopted in this study can be applied to evaluate other types of icon design.

## Introduction

The Covid-19 pandemic broke out in early 2020 and has touched every corner of the globe and every aspect of our lives. Countries around the world have responded to the Covid-19 pandemic crisis by adopting appropriate communication guidelines. In order to accurately and effectively communicate the key points of preparedness to the public, in addition to issuing text-based communication guidelines, government agencies, and health institutions have developed generic communication icons to visualize precautions and measures related to the

**Competing interests:** The authors have declared that no competing interests exist.

COVID-19 pandemic and guidelines for home quarantine. In contrast to traditional print media such as brochures, posters, cartoons, and images, social media is considered a more effective way to promote public health knowledge due to the development of the Internet [1].

Studies have shown that icon design is essential in guiding behavior and decision-making, particularly during health crises [2, 3]. Icons help to facilitate risk communication with the public and enhance understanding of the health risks thereby changing their risky behaviors and effective decision-making [4–6]. Icons have the power to convey complex messages quickly and universally, which can transcend language and culture barriers and can communicate essential health information to diverse populations, including those with low literacy levels or limited language proficiency, promoting inclusiveness and accessibility of health communication [7, 8]. In addition, well-designed icons can create strong visual associations with specific health behaviors or concepts, making them more memorable and recognizable [8, 9]. Icons are widely used in epidemic prevention and guidance, and effective icon design can contain the spread of coronavirus to a certain extent and rapidly improve people's knowledge of coronavirus prevention and control, such as understanding how the virus spreads and avoiding exposure to the coronavirus [3, 10]. Moreover, icons can evoke emotions and influence attitudes and behaviors [11, 12]. For example, icons depicting happy faces or positive imagery associated with healthy behaviors can create a positive emotional response and encourage engagement in those behaviors. Conversely, icons illustrating the consequences of unhealthy behaviors, such as illness or injury, can evoke negative emotions and motivate behavior change. In summary, the impact of iconography on public health communication during a health crisis is multifaceted and includes cognitive, emotional, behavioral, and cultural aspects.

The literature on public health icons or graphic symbol design for health risk communication is limited but growing. According to previous studies, icon physical (external) features such as size, color, dynamism, contrast, clarity, and layout influence users' visual perception and comprehensibility [13–18]. Ryoko Hamaguchi et al. [19], Schneider, Claudia R., et al. [20], Andy J. King [7], and Prasetyo, et al. [3] assessed user perception and understanding of the infographics and icons associated with COVID-19 protection measures, they emphasized that effective use of graphics, icons, and illustrative (photos, illustrations, etc.) can improve health communication during public health crises. Amitabh Verma [21] argues that positively incorporating visual design and emotion in public health iconography can mitigate the negative emotions associated with pandemics with humorous graphics and cartoons. Similar to Amitabh Verma's perspective, Xin Zhao et al. [22] explored Instagram audience's responses to cartoon (graphic medicine) posts related to the COVID-19 pandemic circulating on the platform, and the results of the study demonstrated the viability of graphic medicine as a health communication tool. Additionally, dynamic graphics or images demonstrate multimodal forms of discourse and promotion of healthy behaviors in public health campaign texts, such as Ines Freitas et al. and Esraa Said Abdel Hafez Ahmed Taha et al. analyzed the role of motion graphics in the fight against COVID-19 [23, 24], Tunde Ope-Davies (Opeibi) et al. who discuss web-based public health discursive practices during the Nigerian Coronavirus (COVID-19) pandemic. Public Health Discourse Practices [25], Aisha Saadi Al-Subhi explores semiotically the interactive metadiscourse and meaning construction of posters released during the COVID-19 crisis [26].

These studies assessed the importance and usefulness of COVID-19 icons of different design styles in promoting health information communication. However, assessing the impact of icon design features on icon recognizability and comprehensibility from a user's usage perspective has been neglected, as access to subjective evaluations of icon design is critical to the promotional effectiveness of the icons [17, 27]. In addition, despite the advantages and contributions that icons provide in raising awareness of Covid-19 pandemic prevention, some icons

can be confusing and misleading for users, especially when considering icons with similar contexts (e.g., icons indicating shortness of breath versus dyspnea) [3]. According to previous studies, some icons are difficult to understand [1, 28, 29], especially for older adults [2, 29–32]. Therefore, it is essential to accurately recognize and understand Covid-19 pandemic prevention campaign icons, to understand the impact of the design features of the icons on icon recognition and understanding [33], and, more significantly, to understand subjective preferences such as users' perceptions and attitudes toward the features of these icons [34, 35].

In addition, although icons are widely used as a Covid-19 pandemic visual communication medium, there has been little research on existing Covid-19 pandemic prevention communication icons, particularly a lack of research on users' perceptions and preferences for existing icons. Due to the popularity of the Internet, the function of icons is no longer limited to communication (emphasizing ease of memorization and comprehensibility), and the role of icons has expanded beyond communication to match user preferences [27, 32, 36]. The International Standards Organization (ISO) has emphasized the need to develop icon design principles to ensure visual clarity and subjective preferences to enhance icon recognition and usability [37].

Therefore, this study aims to assess the publicity icons for new crown preventive measures released by health and medical organizations and other official organizations in different countries from the perspective of user needs and perceptions and to screen the icon design features that apply to the publicity of new crown preventive measures. The results of this study will provide a theoretical basis for government agencies, medical organizations, and designers to formulate various public health promotion icon designs.

## Literature review

### Iconic communication

An icon is a widely used and efficient method of communicating information; it is an ideographic graphic symbol that is highly general, effective at communicating information, and simple to recall [38]. Its meaning is created, recognized, and interpreted by associations and memories influenced by brain activity [38]. Iconic communication is the attempt to build cross-language communication systems that completely avoid the use of words and rely solely on pictorial symbols [39]. Iconic communication has a rich history and has been studied extensively in a variety of environments beyond the computer screen, especially in the design of human-computer communication interfaces such as Product Labels, Traffic Signs, Maps, Instructional Manuals, and Computer Icons [17, 40, 41]. Readability and universality are the main characteristics of iconic communication [39, 42]. Iconic communication involves the use of pictographs or icons to convey meaning and information. A pictograph or icon and its message are associated in a many-to-many manner, in other words, a single image might allude to a wide range of thoughts or notions [42]. It is, therefore, a cause of misunderstanding and ambiguity, and the only sure way to reduce the ambiguity of iconic communication is to create codes.

The theoretical foundations of iconic communication are largely based on semiotics and psychology. In the field of design and visual communication, Pierce's trichotomy of icon/index/symbol is the most frequently referred to, it provides a basis for analyzing icons in visual communication design. According to Peirce, a sign as a component of a triadic relation that includes an interpretant (the logic that may be used to connect object and sign), an object (the referent), and a sign or representamen [42]. Indeed, multiple semantic connections between icons and representations can affect icon recognition [42]. Other elements influencing icon recognition include readers' familiarity with the original concept, graphic quality, and representation genre [3, 18, 43]. NEISSER U argues that users' recognition and understanding of

graphical symbols in icons depends on users' prior knowledge base and memory [44], i.e., people's understanding and judgment of sensory-acquired information based on their knowledge and experience [29]. At present, perceptual theories on how to recognize and understand icons/symbols mainly come from Gestalt psychology proposed by Wolfgang Köhler [45]. Gestalt psychology focuses on human vision and sensation, and they found that the determinants of shape are the shapes in the visual field, which in turn are characterized by their outlines or boundaries [46]. Easterby evolved a more systematic theory of perception based on Gestalt psychology, and he argued that, in addition to graphic factors of icons/symbols (outlines, lines, graphic styles), the structural attributes of symbols (including continuity, completeness, symmetry, simplicity, and unity, etc.) are also important factors affecting users' perception and interpretation [47, 48].

Research has shown that icons play a crucial role in communication, as they can create better visual perception and effectively communicate messages [42, 49], different representation strategies can impact how icons are perceived and understood [28]. In the context of health communication, icons have been used to convey information about the health effects of cigarette smoke [50], healthcare symbols help users with wayfinding and medical appointments [51], and a system of symbolic labeling on medicine bottles improves the accuracy of users' medication identification and the effect of container warning labels on users' perceptions of danger [52, 53], and the effectiveness of the visual language of COVID-related signage [3, 21, 22]. These studies indicate that icons can elicit varying degrees of emotional and cognitive elaboration in viewers and that the perceived effects of these icons may also vary based on user cultural differences as well as their design features and representation strategies [12, 33, 39, 49, 54].

## Components and classifications of icon

An icon usually includes a border, background, graphic or symbol elements, and a text label [38, 40, 41]. Although it is not necessary for an icon to contain all of the elements, each icon element can add meaning to the icon; the border can make the icon look more consistent; the background helps to differentiate between the icons; and the graphic and text labels convey the primary meaning of the icon [55].

The most basic classification systems have focused their analysis on the characteristics of the pictorial representation [42]. In the existing literature, most icon categorization is based on the degree of abstraction or figuration of the icon (or graphic). Lodding, K.N. [40], Goonetilleke, Ravindra S. et al. [56], and Wang et al. [57] classified icons into four categories: (1) Image-related, which is a figurative representation of an object or action graphic; and (2) Concept-related, which is an icon attempt to visualize a concept that is not far from, but separate from, a concrete image; (3) Arbitrary icon, which classified according to the degree of similarity of the graphic elements in the icon to their references, arbitrary icons that have no apparent reference to their intended meaning, but can only become meaningful through convention and education; (4) Semi-abstract, which is a combination of all three, figurative icons (image-related) and abstract (concept-related or arbitrary). In addition, if textual elements are combined into icons, two more categories can be added: Textual and Combined icons. Textual icons can be further divided according to whether they contain Word or Abbreviation. Therefore, Chi and Dewi classified icons into seven categories based on this: Image-related, Concept-related, Semi-abstract, Arbitrary, Word, Abbreviation, and combined [33].

## Related studies

Existing literature has explored the importance of user perception of icons and icon design features in iconic communication, and these studies have assessed user perception and

understanding of icon constituents and icon design styles. Stephen Young explored the effects of graphics, colors, and borders of warning icons on increasing attention to warnings, and the results showed that red and yellow were more effective for attention to warning [58]. Dewar and Ells et al. used user reaction time as a valid indicator to assess the perceived quality of traffic signs and showed that icons with graphical symbols as design elements are easier to recognize and understand compared to single textual information, especially in time-limited situations [59, 60]. Similarly, Chi and Dewi investigated the recognition performance of in-vehicle icons in both graphical and textual formats and found that users recognized text-only type icons and figurative icons with a single image more efficiently than graphical and textual combination icons [33]. Overall, images are more universally recognizable than text compared to single textual information [17, 41, 59, 61], and images can avoid problems associated with inadequate reading skills or unfamiliarity with the language and are therefore easier to recognize and remember [33, 62, 63].

User perception assessment of different icon styles has also been the focus of research. Icon style is a summary of the stylistic characteristics of icons from a design perspective, and anthropomorphic icons (figurative) and flat (abstract) icons are the two most common icon design styles in today's icon design field [2]. Studies have shown that anthropomorphic icons are more recognizable than flat icons because they have the shape and appearance of real-world objects [31, 54]. However, not all findings point to the fact that anthropomorphic icons convey information better than flat icons. Liu and Ren Hong et al. used ERP techniques to find that flat icons are concise and efficient compared to anthropomorphic icons, and that users will devote more attention and obtain higher cognitive efficiency [64, 65]. Overall, anthropomorphic icons and flat icons have their advantages and limitations, and designers should design appropriate design styles according to the actual needs of users.

In addition, cultural factors such as demographic factors such as gender, age, and education equally affect users' perception and understanding of icons. Chi and Dewi suggest that participants' age and gender also have an impact on icon style preference, with older participants preferring anthropomorphic icons, while female participants preferring flattened icons [36]. Ying Hu and Jun Liu analyzed the effects of icon style, icon presentation, and age differences on the readability and legibility of waste sorting icons through a subjective measurement experiment, and the study showed that older people preferred anthropomorphic icons, while younger people preferred flattened icons [2]. Similarly, Annie W.Y.'s findings showed that the addition of pictograms in the design of drug packaging icons significantly improved the understanding of drug information among older adults and that less educated older adults had a poorer understanding of drug information [30].

## Evaluation metrics for icon design

In this study, the icon subjectivity evaluation indicators were categorized into three categories based on prior works of literature, as shown in Table 2. The basic considerations for measuring the effectiveness of icon design are the communicative effectiveness of the icon, i.e., whether the icon design accurately conveys its intended meaning [3, 13, 16], the comprehensibility and recognizability of the icon design [3, 66], the degree of clarity and ambiguity with which the icon conveys its meaning [3, 67, 68], the trustworthiness of icons [32].

In addition to communication effectiveness, the visual design perception of icons is significant [17]. Studies have shown that icon designers have personal aesthetic preferences and that aesthetic preferences can also contribute to the understanding and recognition of icons, both from the designers themselves and from the users [27, 28, 69]. Icon design qualities include: boring or interesting [3, 32]; complex or simple [13, 33, 66]; unfamiliar or familiar [13, 33, 66];

not eye-catching or eye-catching [66]; not eye-catching or eye-catching [3, 13]; cluttered or organized [3].

Another metric for evaluating the effectiveness of icon design is Semantic Distance, which measures the closeness of the relationship between a symbol and the content it intends to convey [69]. Usually, this relationship is close and clear, and a function can be directly represented by a figurative graphic or image (e.g., the graphic of a printer directly represents the "print" function). However, in abstract and arbitrary icons, this relationship is less obvious, and the relationship between the function and the graphical symbol is indirect (e.g., the triangles are used to represent "Danger" and "Warning"). In this case, the relationship between what is described in the graphical symbol and the function it represents is much weaker, and the graphical symbol can be understood if the user learns and memorizes it in advance. Therefore, semantic distance indicators include: distant or close [3, 66, 69]; abstract or concrete [3, 13, 33].

## Methods

### Ethics consideration

All procedures performed in studies involving human participants were by the ethical standards of the institutional and/or national research committee and with the 1964 Helsinki Declaration and its later amendments or comparable ethical standards. The experimental protocol obtained informed consent from the participants and did not involve the disclosure of participant privacy, the data were free from commercial interests and intellectual property rights. Therefore, our study followed the exemption from ethical approval by the Ethics Review Committee of Jinan University, China.

### Participants

In this study, questionnaires were collected online using the Questionstar application (https://www.wjx.cn/) and were distributed between September 10 and November 5, 2022. Questionstar is one of the most advanced online survey tools and is also widely used in academic research [70, 71]. The language used for the questionnaire was Chinese by default, and the web link or QR code of the designed questionnaire was forwarded to WeChat groups and friends to achieve online distribution and recovery of the questionnaire. If the participants' questionnaire completion passes the quality control (set the answer time to be not less than 180 seconds), each of them will paid 20RMB after completing the experiment. In addition, a pre-test of 50 samples was conducted before the official test, and the final corrected questionnaire was officially distributed.

This study employed random sampling to select the sample for the study. Since the population of the parent group could not be determined, Bentler and Chou's method of determining sample size was used in this study. According to Bentler and Chou, the sample size is 5–10 times the number of questionnaire items when the parent group of the population is unknown, and the sample size can be enlarged by 20% by considering the number of invalid samples [72]. A total of 223 valid samples were obtained for subsequent statistical analysis after deleting the invalid questionnaires; the recovery rate of the questionnaires was 74.3%, as shown in Table 1. Written informed consent was obtained from participants before questionnaire distribution to ensure confidentiality and voluntary participation before participation. For privacy reasons, the questionnaire did not collect personally identifiable data, such as name, address, and date of birth. In addition, all participants were informed that they may withdraw from the study without any queries or negative consequences.

**Table 1. Demographic characteristics of the participants.**

| Type | Categories | NO. | Percentage% |
|---|---|---|---|
| **Gender** | Male | 102 | 45.7 |
| | Female | 121 | 54.3 |
| | 18–30 | 80 | 35.9 |
| **Age** | 31–45 | 72 | 32.3 |
| | 46–60 | 40 | 17.9 |
| | >60 | 31 | 13.9 |
| **Educational Level** | Junior high school and below | 12 | 5.4 |
| | High School | 26 | 11.7 |
| | College | 44 | 19.7 |
| | Undergraduate | 104 | 46.6 |
| | Postgraduate and above | 37 | 16.6 |
| **Total** | | 223 | 100 |

Table 1 shows the demographic characteristics of the participants. A total of 300 participants were recruited to participate in the study, all of whom were residents of different parts of mainland China. There was a balanced proportion of men and women among the participants, with men (n = 102) accounting for 45.7% and women (n = 121) accounting for 54.3%; the participants were all over 18 years old, and most of the participants were concentrated the younger age group of 18–45 years old (n = 152), which accounted for 68.2% of the participants; the participants' education level was concentrated in college, undergraduate and graduate students (n = 185), which accounted for 82.9%, the good educational background indicates that the participant group has a good cognitive ability of icon design.

## Icons of the study

In this study, the icons came from the COVID-19 pandemic precautionary measures posters or brochures released by 24 governmental agencies or official organizations, and a total of 160 icons were collected and compiled to represent 15 precautionary measures functions respectively. To be more representative and replicable, icons representing different design types, Image-related, Concept-related, Arbitrary, Semi-abstract, and Combined, were purposely selected for this study, excluding single textual and abbreviated types. In addition, textual descriptions were removed from the Combined icons when designing the questionnaire. Due to the high similarity of the icon design styles of some functions, we filtered and deleted these 160 icons, and finally retained 8 icons (120 in total) for each function for the ranking test, as shown in Fig 2.

Considering copyright issues, this study used A,B,C,D,E. . .letters represent the organization and institution from which an icon originated and including in-text citations to these sources:**A** (Guangzhou Health Commission, China) [73]; **B** (China Center for Disease Control and Prevention (CDC)) [74]; **C**(National Health Commission of the People's Republic of China) [75]; **D** (Office of the Foreign Affairs Working Committee of the Hunan Provincial Committee of the Communist Party of China, China) [76]; **E** (South Carolina Departments of Environmental Services (SCDES)) [77]; **F** (Centers for Disease Control and Prevention CDC, USA) [78]; **G** (Australian Commission on Safety and Quality in Health Care) [79]; **H** (Ohio State Department of Health) [80]; **I** (Pennsylvania Department of Health) [81]; **J** (Ministry of Healthand Family Welfare Government of India) [82]; **K** (www.MEDIUM.com) [83]; **L** (WHO Africa) [84]; **M** (Floridian State Government Department of Health) [85]; **N** (Asanti Africa Foundation) [86]; **O** (Thailand Convention and Exhibition Bureau) [87]; **P**

(Government of Ireland Department of Health) [88]; **Q** (World Health Organizition) [89]; **R** (MDPI Article Fig 2. Preventive measures for the COVID-19 pandemic.) [90]; **S** (Japan National Tourism Organization) [91]; **T** (Tennessee Department of Health) [92]; **U** (Korea's Central Department of Epidemic Prevention and Countermeasures) [93]; **V** (Ministry of Public Health (MOPH), Qatar) [94]; **W** (www.MEDIUM.com) [95]; **X** (www.shutterstock.com) [96].

## Ranking test

The first phase of this study was the Appropriateness Ranking Test (ART), in which participants were tasked with ranking icons of the same function type according to the principle of relative appropriateness [97]. The Ranking Test was conducted using the Questionnaire Star App,and the test data files in (S1 Table). The icons were coded as "a, b, c, d, e, f, g, and h". The icon size was 120 pixels by 120 pixels (4.2 cm by 4.2 cm). The average composite score for each icon is automatically calculated by the Questionnaire Star system, which reflects the overall ranking of the options, with higher scores indicating a higher overall ranking, as shown in Fig 2. The formula for calculating the score is as follows:

*Average composite score = (Σ frequency × weight) / number of times the question is filled in.*

If three options are involved in the sorting, the first position in the ranking of the weight value is 3, the second position weight value is 2, and the third position weight value is 1. For example, a total of 12 times a question was filled out, Option A was selected and ranked in the first position 2 times, the second position 4 times, the third position 6 times, and the average composite score of Option A = $(2 × 3 + 4 × 2 + 6 × 1) / 12 = 1.67$ points.

## Subjective rating test

Subjective Rating Tests (SRT) are used in almost every aspect of ergonomics research and practice, are easy to administer, and are more scientific and sensitive than objective measurements [98]. In addition, subjective rating tests are one of the most effective methods for assessing the comprehensibility of text and graphic symbols [13, 66, 99], which can systematically assess the impact of graphical symbol features on User Performance [69]. In this study, three subjective design features for evaluating icons were summarized based on previous research: (1) communication effectiveness [3, 66]; (2) Visual Design Perception [3, 66]; and(3)Semantic distance [3, 66, 69]. In this phase, participants were asked to rate the icons that ranked 1st in the previous phase of the ranking test based on the subjective design features. To compare the types of icon styles preferred by users, we classified the icons into two categories: (1) figurative icons, i.e., icons that contain specific characters, actions, and scenes, which are more concrete and vivid, and have a sense of familiarity; and (2) abstract icons, i.e., generalized and simplified icons, which are more concise and abstract. In this study, three groups of icons with the same functions, which can represent the above two styles were selected for testing to examine whether there are significant differences between these two groups of icons on the 12 semantic scales (S2 Table).

This study phase was conducted through the Questionnaire Star App and was rated using a 7-level semantic differential scale, as shown in Fig 1. Table 2 demonstrates the 12 sets of semantic vocabularies in the three subjective design features collected and organized from the relevant literature. The quantitative analysis of perceptual semantic vocabulary is realized through the Semantic Difference Method, which transforms multiple perceptual vocabulary variables into preferred composite variables, and ultimately, ambiguous and unknown

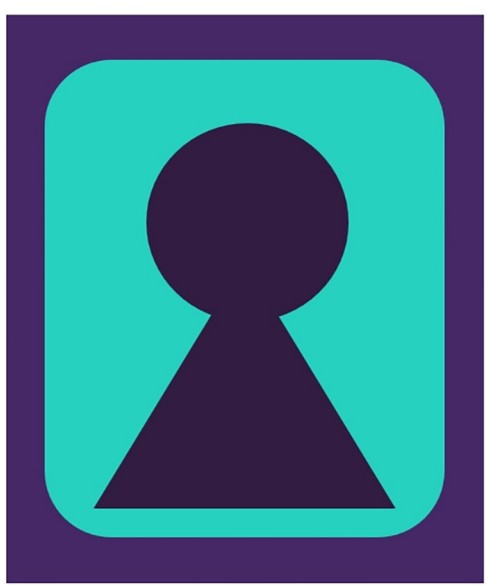

**Fig 1.**

perceptual demands that are difficult to analyze can be transformed into icon design elements [71, 100–102]. This approach was based on our research goal of assessing users' perceptions and preferences for each icon design feature and the strength of these feature aggregations.

**Table 2. Subjective design features and perceptual semantic vocabulary.**

| Subjective Design Features | Perceptual Semantic Vocabulary | Literature sources |
|---|---|---|
| **Communication Effectiveness** | Unrecognizable-Recognizable | Liu and Ho (2012);Prasetyo et al. (2021);Lin (1992);Chi et al. (2019);Urbano, Inês Cunha Vaz Pereira et al. (2022);Zou et al. (2019);Lee et al. (2014) |
| | Fuzzy—Clear | |
| | Questionable—Reliable | |
| | Ineffective—Effective | |
| | | |
| **Visual Design Perception** | Ugly—Beautiful | S. J. McDougall et al. (1999);Jylhä and Hamari (2020);Collaud et al. (2022; Lin);Liu and Ho (2012);Prasetyo et al. (2021) |
| | Boring—Interesting | |
| | Complex—Simple | |
| | Unfamiliar—Familiar | |
| | Not Eye-Catching—Eye-Catching | |
| | Cluttered—Organized | |
| **Semantic Distance** | Distant—Close | Liu and Ho (2012);Lin (1992);Prasetyo et al. (2021);S. J. McDougall et al. (1999);Ou and Liu (2012) |
| | Abstract—Concrete | |

Participants evaluated each icon on a perceptual vocabulary semantic difference scale, and the closer to the left or right side of the semantic difference scale they chose, the more they perceived the icon to be a better fit for this perceptual vocabulary.

## Statistical analysis

SPSS 24 software was used to analyze the data in this study. The questionnaire survey combined with the semantic differential method to transform the data from users' perceptual evaluations of the sample icons, and the magnitude of the values intuitively indicated the degree of relationship between the sample icons and the design features [35, 103]. A one-sample t-test was performed on the 12 semantic scales to determine how they differed from the mean. Spearman correlation analysis was used to derive correlation (two-tailed) results for bipolar perceptual words. Examining the correlations between perceptual semantic words helps to understand the correlations between icon design features. Then, To compare whether users prefer anthropomorphic or images that contain a specific scene i.e. more figurative, Friedman's test (S3 Table) was used to determine the real differences between the 12 semantic scales, which can help scholars to design or select icons related to the COVID-19 prevention measures. In addition, factor analysis [104, 105] was used to downscale the icon design features to determine the perceived effects and subjective preferences of users on the perceptual semantic vocabulary of the sample icons, which can help government departments and designers refer to the icon design characteristics of Covid-19 prevention measures.

## Results

### Results of the icon ranking test

Fig 2 shows the results of all the icon ranking tests. The results show that most participants preferred image-based and combined icons. Among the top 15 icon function types, all of them were image-based, and 8 of them related to cartoon characters' images received high ratings, which were 01-Wash Hands, 02-Ventilate Diligently, 03-Wear A Mask, 05-No Gathering, 07-Clean and Hygienic, 10-Keep Hands Off Your Face,12-Check Temperature and 15-Getting Vaccinated. These icons used vivid cartoon graphics depicting characters and actions, such as washing hands, opening windows, wearing masks, and engaging in hygiene. The two icons, 08 —Exercise and 11—Cough Etiquette, use silhouettes of characters and are the most concise and single composition of all the icons, with relatively low icon ratings. 03-Wear A Mask, 04-Keep Social Distance, 05- No Gathering, 06-Stay At Home, 10-Keep Hands Off Your Face, 13-Travel Restrictions, And 14-Seek Medical Help are combination icons that contain figurative graphics such as a person or scene, abstract symbols (prohibited ⊘ and permitted ✓ symbols), and textual information that are helpful for users to understand. It is worth noting that some of the top-ranked combo icons were rated lower than the other top-ranked icons, probably because they used a more minimalist, generalized graphic design that was not realistic and interesting to the user, e.g., 04- Keep Social Distance, 06-Staying at Home, 13-Travel Restrictions, and 14-Seeking Medical Help. However, looking at the ranking of all the icon samples, the combinatorial icons still received a relatively high ranking.

### Results of the icon subjective rating test

#### Reliability statistics of icon design features

The reliability statistics under each dimension variable are shown in Table 3. The Cronbach's $\alpha$ is used to assess the internal consistency of the measurement instrument, which usually ranges from 0.7 to 1, with higher values indicating better reliability. In addition, the inter-item

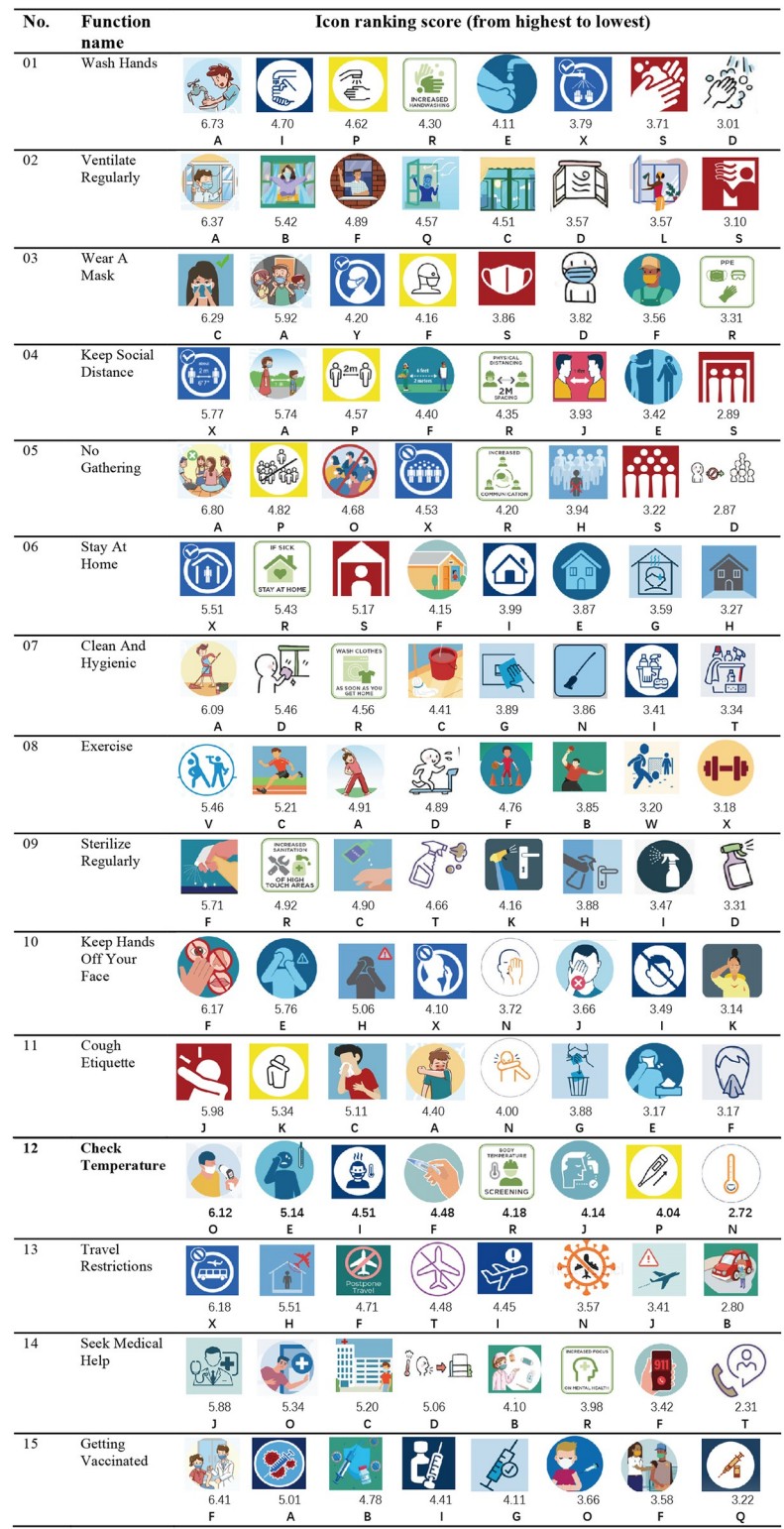

**Fig 2.**

**Table 3. Reliability statistics for three dimensions of subjective design features.**

| Dimensions | Item | CITC | Deleted items α | Cronbach α |
|---|---|---|---|---|
| **Communication Effectiveness** | Recognizable | 0.771 | 0.984 | 0.96 |
| | Clear | 0.957 | 0.931 | |
| | Reliable | 0.952 | 0.933 | |
| | Effective | 0.934 | 0.938 | |
| **Visual Design Perception** | Beautiful | 0.947 | 0.985 | 0.987 |
| | Interesting | 0.957 | 0.984 | |
| | Simple | 0.945 | 0.985 | |
| | Familiar | 0.958 | 0.984 | |
| | Eye-Catching | 0.969 | 0.983 | |
| | Organized | 0.959 | 0.984 | |
| **Semantic Distance** | Close | 0.767 | - | 0.868 |
| | Concrete | 0.767 | - | |

correlation coefficient (CITC value) for each indicator was also counted to assess the correlation between the indicators. The results of the analysis show that the CITC value of each indicator is greater than 0.7, which indicates a good correlation between them, and also indicates a good level of reliability. Overall, the Cronbach's $\alpha$ of Communication Effectiveness, Visual Design Perception, and Semantic Distance are 0.96, 0.987, 0.868, which, combined with the CITC values of the indicators, indicates that the data have a high quality of reliability, which can provide a reliable basis for further research and analysis.

## Descriptive statistics of icon design features

Table 4 shows the characteristics of the data distribution for each indicator. The sample size is 223, and for each adjective variable, the minimum, maximum, mean, and standard deviation are given. As can be seen from the results, there is a wide range of values for each variable, and the Mean and Standard Deviation indicate the degree of variability in the sample. For example, Recognizable has a mean of 82.7713 and a standard deviation of 15.85312, indicating that the observations in the sample are relatively concentrated around this mean, but there is some degree of dispersion. Similarly, other variables such as Clear, Reliable, and Effective show a similar picture.

**Table 4. Descriptive statistics.**

| Dimensions | Item | N | Minimum | Maximum | M | SD |
|---|---|---|---|---|---|---|
| **Communication** | Recognizable | 223 | 33 | 105 | 82.7713 | 15.85312 |
| **Effectiveness** | Clear | 223 | 35 | 105 | 79.3901 | 16.6578 |
| | Reliable | 223 | 16 | 105 | 76.4081 | 17.56248 |
| | Effective | 223 | 32 | 105 | 78.0269 | 16.78344 |
| **Visual Design** | Beautiful | 223 | 19 | 105 | 77.4484 | 17.52311 |
| **Perception** | Interesting | 223 | 33 | 105 | 79.417 | 16.53146 |
| | Simple | 223 | 29 | 105 | 79.6413 | 17.28866 |
| | Familiar | 223 | 28 | 105 | 78.8161 | 16.68919 |
| | Eye-Catching | 223 | 23 | 105 | 77.8206 | 18.46409 |
| | Organized | 223 | 28 | 105 | 79.7758 | 17.57747 |
| **Semantic** | Close | 223 | 28 | 105 | 81.5336 | 16.48962 |
| **Distance** | Concrete | 223 | 31 | 105 | 85.5605 | 16.8428 |
| | Valid N (listwise) | 223 | | | | |

**Table 5. One-sample t-test of 15 icon semantic vocabularies.**

| Dimensions | Item | Minimum | Maximum | M | SD | t | P |
|---|---|---|---|---|---|---|---|
| Communication Effectiveness | Recognizable | 33 | 105 | 82.771 | 15.853 | 28.515 | 0.000** |
| | Clear | 35 | 105 | 79.39 | 16.658 | 24.106 | 0.000** |
| | Reliable | 33 | 105 | 76.408 | 17.562 | 20.329 | 0.000** |
| | Effective | 32 | 105 | 78.027 | 16.783 | 22.713 | 0.000** |
| Visual Design | Beautiful | 19 | 105 | 77.448 | 17.523 | 21.261 | 0.000** |
| Perception | Interesting | 16 | 105 | 79.417 | 16.531 | 24.315 | 0.000** |
| | Simple | 29 | 105 | 79.641 | 17.289 | 23.443 | 0.000** |
| | Familiar | 28 | 105 | 78.816 | 16.689 | 23.547 | 0.000** |
| | Eye-Catching | 23 | 105 | 77.821 | 18.464 | 20.479 | 0.000** |
| | Organized | 28 | 105 | 79.776 | 17.577 | 23.173 | 0.000** |
| Semantic | Close | 28 | 105 | 81.534 | 16.49 | 26.293 | 0.000** |
| Distance | Concrete | 31 | 105 | 85.561 | 16.843 | 29.312 | 0.000** |

* p<0.05

** p<0.01

## T-test of 15 icon semantic vocabularies

Since each variable is obtained by combining the 15 icon semantic vocabularies, each option is assigned a value of 1–7, so when all 15 icons are selected as 7, the total score is 105, and the median of the criterion is 52,5, so the reference value is set to be 52.5 when conducting the one-sample t-test. As can be seen from Table 5, the mean values of Recognizable, Clear, Reliable, Effective, Beautiful, Interesting, Simple, Familiar, Eye-Catching, Organised, Close, and Concrete were significantly higher than 52.5, and all of them show significance (p<0.05), which indicating that the mean values of the 12 semantic vocabularies are statistically different from the median value of 52.5.

In addition, focusing on the effect size metric, i.e., Cohen's d value can be further tested for the difference between the mean score and the median (52.5) for each semantic vocabulary. Table 6 shows that Cohen's d values for most of the adjectives ranged from 1.4 to 1.9 and none of the 95% Confidence Intervals for the differences contained zero values, indicating that all of the icons tested possessed Recognizable, Clear, Reliable, Effective, Beautiful, Interesting,

**Table 6. Efficacy analyses (median 52.5).**

| Item | Mean | DV | DV95% CI | SD | Cohen's d |
|---|---|---|---|---|---|
| Recognizable | 82.771 | 30.271 | 28.179 ~ 32.363 | 15.853 | **1.909** |
| Clear | 79.39 | 26.89 | 24.692 ~ 29.088 | 16.658 | 1.614 |
| Reliable | 76.408 | 23.908 | 21.590 ~ 26.226 | 17.562 | 1.361 |
| Effective | 78.027 | 25.527 | 23.312 ~ 27.742 | 16.783 | 1.521 |
| Beautiful | 77.448 | 24.948 | 22.636 ~ 27.261 | 17.523 | 1.424 |
| Interesting | 79.417 | 26.917 | 24.735 ~ 29.099 | 16.531 | **1.628** |
| Simple | 79.641 | 27.141 | 24.860 ~ 29.423 | 17.289 | 1.57 |
| Familiar | 78.816 | 26.316 | 24.114 ~ 28.519 | 16.689 | 1.577 |
| Eye-Catching | 77.821 | 25.321 | 22.884 ~ 27.757 | 18.464 | 1.371 |
| Organized | 79.776 | 27.276 | 24.956 ~ 29.595 | 17.577 | 1.552 |
| Close | 81.534 | 29.034 | 26.858 ~ 31.210 | 16.49 | **1.761** |
| Concrete | 85.561 | 33.061 | 30.838 ~ 35.283 | 16.843 | **1.963** |

**Table 7. Spearman correlation results between 12 semantic scales.**

|  |  | 1 | 2 | 3 | 4 | 5 | 6 | 7 | 8 | 9 | 10 | 11 | 12 |
|---|---|---|---|---|---|---|---|---|---|---|---|---|---|
| 1 | **Recognizable** | 1 |  |  |  |  |  |  |  |  |  |  |  |
| 2 | **Clear** | .747** | 1 |  |  |  |  |  |  |  |  |  |  |
| 3 | **Reliable** | .724** | **.933**** | 1 |  |  |  |  |  |  |  |  |  |
| 4 | **Effective** | .701** | .916** | **.939**** | 1 |  |  |  |  |  |  |  |  |
| 5 | **Beautiful** | .688** | .866** | .891** | .888** | 1 |  |  |  |  |  |  |  |
| 6 | **Interesting** | .668** | .886** | .916** | .915** | **.941**** | 1 |  |  |  |  |  |  |
| 7 | **Simple** | .647** | .864** | .884** | .871** | .880** | .909** | 1 |  |  |  |  |  |
| 8 | **Familiar** | .678** | .881** | .913** | .898** | .886** | .913** | .896** | 1 |  |  |  |  |
| 9 | **Eye-Catching** | .674** | .898** | .912** | .898** | .893** | **.928**** | .918** | **.939**** | 1 |  |  |  |
| 10 | **Organized** | .676** | .888** | .902** | .878** | .866** | .888** | .903** | .916** | **.940**** | 1 |  |  |
| 11 | **Close** | .722** | .872** | .879** | .891** | .849** | .856** | .867** | .875** | .889** | .894** | 1 |  |
| 12 | **Concrete** | **.934**** | .718** | .711** | .683** | .717** | .702** | .699** | .723** | .741** | .701** | .738** | 1 |

**. Correlation is significant at the 0.01 level (two-tailed).

Simple, Familiar, Eye-Catching, Organised, Close, Concrete. The largest effect size was found for Concrete (1.963), which means that the adjective Concrete had the most significant difference in ratings from the reference value, followed by Recognizable (1.909), Close (1.761), and Interesting (1.628).

## Spearman correlations between the 12 semantic scales

Table 7 demonstrates the results of the Spearman correlations (two-tailed) between the 12 semantic scales. The analysis results show that there are significant inter-correlations between the 12 semantic scales. Among them, Recognizable and Concrete, Clear and Reliable, Reliable and Effective, Beautiful and Interesting, Interesting and Eye-Catching, Familiar and Eye-Catching, Eye-Catching and Organised had the highest correlation coefficients, i.e., 0.934, 0.933, 0.939, 0.941, 0.928, 0.939, 0.940 respectively.

## Friedman's test in chi-square statistic

In this study, 3 groups of icons with the same function that can represent the above two styles respectively were selected to test whether there is a significant difference between these two groups of icons on the 12 semantic scales by Friedman's test. Table 8 shows the results of the

**Table 8. Friedman's test in chi-square statistic ($\chi 2$).**

| Group | Test Statistics | Recognizable | Clear | Reliable | Effective | Beautiful | Interesting | Simple | Familiar | Eye-Catching | Organized | Close | Concrete |
|---|---|---|---|---|---|---|---|---|---|---|---|---|---|
| **Group A** | Chi-Square | 17.785 | 0.73 | 7.903 | 2.429 | 2.674 | 20.632 | 13.714 | 12.082 | 13.591 | 2.613 | 11.893 | 17.294 |
|  | Asymptotic significance | 0.000** | 0.787 | 0.005 | 0.119 | 0.102 | 0.000** | 0.000** | 0.001* | 0.000** | 0.106 | 0.001* | 0.000** |
| **Group B** | Chi-Square | 13.703 | 11.61 | 8.393 | 16.696 | 7.723 | 22.73 | 7.42 | 13.634 | 14.029 | 0.247 | 15.027 | 24.5 |
|  | Asymptotic significance | 0.000** | 0.001* | 0.004* | 0.000** | 0.005 | 0.000** | 0.006 | 0.000** | 0.000** | 0.619 | 0.000** | 0.000** |
| **Group C** | Chi-Square | 20.492 | 0.114 | 12.27 | 21.154 | 7.605 | 18.333 | 4.31 | 10.314 | 13.113 | 4.721 | 12.094 | 16.79 |
|  | Asymptotic significance | 0.000** | 0.735 | 0.000** | 0.000** | 0.006 | 0.000** | 0.38 | 0.000** | 0.000** | 0.3 | 0.001* | 0.000** |

**. Chi-square statistic is significant at the 0.01 level.

*. Chi-square statistic is significant at the 0.05 level (two-tailed).

variance and significance of all Friedman's tests for these three groups of icons on each semantic scale. Based on this table, we can see that there are significant differences between the 2 different styles of icons for the same function on different semantic scales. Taken together, the 3 groups of icons show significant differences in the semantic scales Recognizable, Reliable, Effective, Interesting, Familiar, Eye-Catching, Close, and Concrete, which suggests that, compared to the abstracted icons, the figurative icons in the Recognizable, Reliable, Effective, Interesting, Familiar, Eye-Catching, Close and Concrete features are more pronounced, the result is similar to the results from T-test (Table 6) and the Sperman correlation analysis (Table 7). The differences in Clear, Beautiful, Simple, and Organized are not significant, indicating that users perceive abstract icons to be just as good as figurative icons, with or without Clear, Beautiful, Simple, and Organized design features.

## Factor analysis of icon design features

Factor analysis and principal component analysis were performed to screen keywords for icon design features further. The data were subjected to dimensionality reduction, and the factors were rotated to reduce the number of variables to explain the semantic vocabulary of subjective evaluation of icons with the least number of factors. SPSS analysis results showed that the sample data statistic KMO = 0.958, $P < 0.05$, which indicates the data was suitable for principal component analysis.

The common factor variance indicates the interpretation of the extracted common factor on the original information of the variable, and the degree it reaches, the closer the extracted value is to 1, which indicates that the factor analysis of the variable is more effective. As can be seen from Table 9, the extracted values range from 0.945 to 0.979, of which "Concrete," "Interesting," and "Concise" are the three semantic vocabularies with the highest scores, 0.979, 0.978, and 0.975 respectively, which indicates that these are the three icon design qualities that users value most.

The number of principal components can be determined from the total variance explained, as shown in Table 10. There are two components with eigenvalues > 1, principal component 1 has a variance contribution value of 65.2%, and principal component 1 has a variance contribution value of 25.353%, and the two components can explain a total of 93.552% of the variance of the original variable, so these two components were extracted as principal components. From the rotated component matrix (see Table 11), we can get the ordering of the 12 groups of semantic vocabularies; the first three in principal component 1 are

**Table 9. Common factor variance.**

| Vocabularies | Initial Values | Extraction |
|---|---|---|
| Recognizable | 1.000 | 0.973 |
| Clear | 1.000 | 0.960 |
| Reliable | 1.000 | 0.966 |
| Effective | 1.000 | 0.962 |
| Beautiful | 1.000 | 0.978 |
| Interesting | 1.000 | 0.975 |
| Simple | 1.000 | 0.948 |
| Familiar | 1.000 | 0.951 |
| Eye-Catching | 1.000 | 0.953 |
| Organized | 1.000 | 0.957 |
| Close | 1.000 | 0.945 |
| Concrete | 1.000 | 0.979 |

**Table 10. Total explanation of variance.**

| Component | Initial eigenvalue | | | Rotational load sum of squares | | |
|:---:|:---:|:---:|:---:|:---:|:---:|:---:|
| | Total | Percentage of variance | Cumulative % | Total | Percentage of variance | Cumulative % |
| 1 | 10.599 | 88.324 | 88.324 | 7.824 | 65.200 | 65.200 |
| 2 | 0.627 | 5.228 | 93.552 | 3.402 | 28.353 | 93.552 |
| 3 | 0.170 | 1.418 | 94.970 | | | |
| 4 | 0.152 | 1.265 | 96.235 | | | |
| 5 | 0.093 | 0.771 | 97.006 | | | |
| 6 | 0.079 | 0.661 | 97.667 | | | |
| 7 | 0.073 | 0.609 | 98.276 | | | |
| 8 | 0.056 | 0.470 | 98.746 | | | |
| 9 | 0.048 | 0.397 | 99.143 | | | |
| 10 | 0.037 | 0.311 | 99.454 | | | |
| 11 | 0.035 | 0.288 | 99.742 | | | |
| 12 | 0.031 | 0.258 | 100.000 | | | |

"Interesting," "Concise," and "Familiar," and in principal component 2 are "Recognizable," "Concrete" and "Close(semantic distance)" respectively.

## Discussion

### User preferences for icon design styles

All of the top 15 icon function types are image-based, with 8 icons related to cartoon characters, such as "01-Wash Hands", receiving high ratings. These icons use figures or cartoon drawings to depict characters and actions, such as washing hands, opening windows, putting on masks, and doing hygiene; 7 icons, such as "04-Keep Social Distance", use silhouettes of characters, which is the most concise and unitary form of composition of all the icons, and the icon scores are relatively lower. From the ranking results, users seem to prefer icons that contain specific character images, actions, and scenes. These icons have a higher degree of restoration

**Table 11. Rotated component matrix[a].**

| Vocabularies | Component | |
|:---:|:---:|:---:|
| | 1 | 2 |
| Recognizable | 0.409 | 0.884 |
| Clear | 0.828 | 0.495 |
| Reliable | 0.863 | 0.458 |
| Effective | 0.86 | 0.415 |
| Beautiful | 0.879 | 0.42 |
| Interesting | 0.882 | 0.395 |
| Simple | 0.88 | 0.385 |
| Familiar | 0.87 | 0.427 |
| Eye-Catching | 0.874 | 0.43 |
| Organized | 0.871 | 0.43 |
| Close | 0.83 | 0.668 |
| Concrete | 0.44 | 0.862 |

Rotation method: the Kaiser normalized maximum variance method. The rotation
[a] has converged after three iterations.

of the original references during the design process, with added, specific details that users are familiar with, and a cartoon drawing style that makes the icons more interesting. To further confirm the results of the icon ranking, this study categorized the icons into figurative icons and abstract icons for comparative testing. Three groups of icons with the same function and representing the two styles were selected to test whether there are significant differences between these two groups of icons on the 12 semantic scales. According to the results of Friedman's test, it is inferred that compared with abstract icons, figurative icons have higher scores in Recognizable, Effective, Interesting, Familiar, Eye-Catching, Close, and Concrete, which are the main factors determining users' preferences. This is also a major factor in determining user preference, as these features are more helpful in enhancing the recognizability and communicative effect of the icon.

For some icons at the bottom of the ranking test, the overall rating is relatively low even though they are all image-related, and this may be because the images in these icons are drawn as silhouettes. The highly simplified icons lose many vivid details despite their cleaner visual effect. This is consistent with previous findings that users prefer figurative icons (containing images that are familiar or concrete to the user) and that users are more likely to recognize graphical symbols that represent items that are more accessible in daily life [63, 69, 106], as concrete images or symbols tend to be more visually appealing [28, 107]. Since familiar and specific details such as people, things, or scenes are helpful for users to understand enhances the efficiency of user recognition [108, 109]. Therefore, the more specific the icon, the closer the semantic distance, the faster and more accurately the user can respond and influence the user's aesthetic preferences for icon design [28, 41]. Also, adding forbidden(/)and permitted(/) symbols to icons can make them more straightforward, accurate, and preferred by users. Moreover, the results of this icon ranking suggest that the color of the icon is also an essential factor influencing user preference, with icons that are grey, dark, or cluttered receiving lower ranking scores.

## Design features of user preferences for icons

Spearman's correlation and factor analyses showed that the main design features that influence users' perceptions and preferences for the COVID-19 preventive measures icons are "Interesting," "Simple," "Familiar, "Recognizable," "Concrete" and "Close(semantic distance)." In previous studies, "familiarity," "concreteness" and "Accuracy of semantic depiction" of graphic symbols have been identified as the main factors affecting users' recognition and understanding of icons, while simplicity and meaningfulness have been identified as less important factors in symbol design [13, 69]. Research has shown that familiar, figurative graphic symbols are more accurate and easier to recognize and understand than unfamiliar, abstract graphic symbols [3, 13, 33, 69], which may be because the relationship between the function and the graphic symbol is direct in familiar, and concrete icons [69]. However, in this study, the icon design feature of "Simple" is as important as "Familiar" and "Concrete" to users. The simpler and more specific the icon is, the closer the familiarity and semantic distance, the easier it is for users to perceive it, and the more efficiently they recognize the icon [13, 16, 43]. It is also worth noting that the icon "Interesting" is considered the most favorite icon design feature of users, but "Interesting" is a difficult metric to evaluate, mainly related to the icon shape and design style. Chi and Dewi studied seven styles of bright home icons for living rooms and bathrooms through sorting and comprehensibility tests, and the results showed that users' preferences for flat and anthropomorphic icons were relatively balanced, while their preferences for semi-anthropomorphic icons were relatively low [36]. Some studies have shown that anthropomorphic icons are easier to recognize than flat icons because they have the shape and

appearance of real-world objects [31, 54]. However, in this study, users found figurative or comic book-style icons more interesting than flat and silhouette form icons. Cartoon characters are more vivid and interesting, as well as containing richer details and colors, which are to some extent more aesthetically pleasing to users.

## Conclusions

This study assessed users' perceptions and preferences for COVID-19 preventive measures promotional icons through appropriateness ranking tests and semantic differential methods. Current findings suggest that the images presented in icons are key to icon perception and preference. The results of the study showed that "Interesting," "Simple," "Familiar, "Recognizable," "Concrete," and "Close(semantic distance)" are the key design features that influence the user's perception of icon quality, and the user's preference for icons also relies more on these features. However, government departments and public health organizations prioritize simple, general, and standardized icon design styles for seriousness, scientific, and communicative reasons when developing promotional icon programs, which circumvents the idea of fun and vivid graphic design. However, this may ignore users' aesthetic preferences to a certain extent and pull away the communication distance between publicity icons and the public, which leads to low favorability and insufficient understanding of the icons and may lead to less-than-ideal publicity effects.

The results of this study can provide government departments, public health organizations, and designers with ideas and design directions for improving related publicity icons. In the future, the following design features can be considered to improve the related icons: (1) Enhance the interestingness of icon design. Developing icons based on the principles of fun, relaxation, and familiarity is conducive to increasing the affinity and vitality of icons; (2) Using a figurative and realistic icon design style. Reduce the use of minimalist and abstract images; use more combined icons (with text information and permitted/prohibited symbols) to further increase the user's perception and understanding of the icon; (3) Develop an appropriate icon design color scheme. Avoid using black and white, gray, and other low hues or cluttered color schemes; choose a more saturated or eye-catching color scheme for the design to make the icon more visually attractive and approachable. In addition, a unified color scheme helps to establish icon standardization and can enhance public trust in official institutions and organizations.

This study also has some limitations. Due to time and financial constraints, a larger valid sample could not be collected. Therefore, the findings of this study only represent the attitudes and opinions of this population segment. Future researchers may consider increasing the number of samples to obtain more comprehensive results. There is a need to consider the differences in the perception and comprehension of icons by demographic variables such as gender, age, education level, and field of specialization. Secondly, the icons in this study were from 24 countries or regions; however, cross-cultural differences in the understanding of icons were not discussed in this study, and future research could increase cross-cultural comprehension validity. As age increases, design research for the elderly population will be a hot research topic in the future; studies have shown that there are significant differences in the ability to understand symbols between young and older adults [2, 13, 32]. In addition, this study focuses on icon perception and preference testing from the subjective perspective of users. Although the test itself has some advantages, it needs to take into account the bias of users' subjective opinions because the way icons are presented also affects users' perceptions, and complex icons are gradually perceived as more straightforward as subjects learn and familiarize themselves with the icons [99, 110].

## Supporting information

**S1 Table. Ranking test data.**
(XLSX)

**S2 Table. Subjective rating test data.**
(XLSX)

**S3 Table. Friedman test data.**
(XLSX)

## Acknowledgments

Firstly, we would like to thank all of the participants who took part in the study- we are grateful for your time. We would also like to thank Prof. Chunhua Lin, Prof. Qiong Liao, and Prof. Shuicai Qiu for their contributions to this study.

## Author Contributions

**Data curation:** Qiong Liao, Shuicai Qiu.

**Formal analysis:** Qiong Liao, Shuicai Qiu.

**Methodology:** Deng Lujie.

**Software:** Deng Lujie.

**Supervision:** Chunhua Lin.

**Writing – original draft:** Deng Lujie.

**Writing – review & editing:** Chunhua Lin, Qiong Liao.

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
