## [Decision Letter · Decision Letter 0]

9 Jan 2024

PONE-D-23-31275The effects of icon design features on user perception and preference: A case study of icons for Covid-19PLOS ONE

Dear Dr. LUJIE,

Thank you for submitting your manuscript to PLOS ONE. After careful consideration, we feel that it has merit but does not fully meet PLOS ONE’s publication criteria as it currently stands. Therefore, we invite you to submit a revised version of the manuscript that addresses the points raised during the review process.

The evaluation performed by two excellent reviewers and myself has concluded  that your texts requires a major revision on methodological aspects which might improve your results and its scientific contribution.

Please find below the specific items that need to be addressed.

We look forward to receiving your revised manuscript.

Kind regards,

Jorge Abelardo Falcón‐Lezama, PhD

Guest Editor

PLOS ONE

Journal Requirements:

4. We note that Figures 1, 2, 3, 4, 5 and table 2 in your submission contain copyrighted images. All PLOS content is published under the Creative Commons Attribution License (CC BY 4.0), which means that the manuscript, images, and Supporting Information files will be freely available online, and any third party is permitted to access, download, copy, distribute, and use these materials in any way, even commercially, with proper attribution. For more information, see our copyright guidelines: http://journals.plos.org/plosone/s/licenses-and-copyright.

a. You may seek permission from the original copyright holder of Figures 1, 2, 3, 4, 5 and table 2 to publish the content specifically under the CC BY 4.0 license. 

5. Please include a copy of Tables 3, 4, 5, 6 and 7 which you refer to in your text on pages 10, 20, 21 and 22.

Reviewers' comments:

Reviewer's Responses to Questions

**Comments to the Author**

1. Is the manuscript technically sound, and do the data support the conclusions?

Reviewer #1: No

Reviewer #2: Partly

2. Has the statistical analysis been performed appropriately and rigorously? 

Reviewer #1: No

Reviewer #2: No

3. Have the authors made all data underlying the findings in their manuscript fully available?

Reviewer #1: No

Reviewer #2: Yes

4. Is the manuscript presented in an intelligible fashion and written in standard English?

Reviewer #1: No

Reviewer #2: Yes

5. Review Comments to the Author

Reviewer #1: PONE-D-23-31275

The study entitled: “The effects of icon design features on user perception and preference: A case study of icons for Covid-19” is an interesting study with some good stimulus material and good data collection. However, there are number of problems that would prevent publication of this study in Plos One. The major problem is a hodge podge of statistics that are in search of the question. The author could have and should have been able to determine which icons were most effective on their three dimensions within the group (e.g. for example for handwashing or mask wearing). That would have been useful information. So, I will begin with the major problems with the results and then discuss other problems.

RESULTS

The results need a major overhaul. The result lack statistical tests and most of the results are subjective and obtained through “eyeballing.”

1. The authors could have done a factor analysis and computed reliability coefficients for each of the 3 variables: communication effectiveness, design quality, and semantic distance but this is missing. The authors have psychometrically valid data for each of these dimensions but they do not use it. Since each of these scales contain multiple semantic differential items that represent the dimension an internal reliability coefficient is necessary and easy to compute. From that point some meaningful statistical tests could have been conducted and reported.

2. We need to know if results for each dimension were aggregated and divided by the number of items that comprise the dimension or what. I cannot find scores for each dimension. This needs major revision and clarification.

3. Just reporting the top 15 icon function types is subjective; were these significantly different?

4. What does it tell is that some function types are better than others? Does it mean that we should tell people to wash hands but not check their temperature? The research question needs to be posed and then systematically answered.

5. We need statistical evidence to back the authors’ assertion that: “it can be initially determined that users prefer icons that contain specific character images, actions.”

6. Similarly, the authors assert that “It is suggested that users prefer icons associated with images when real or illustrated more concretely.” What statistical evidence supports this assertion?

7. What would be interesting would be to ascertain for each type of icon which was significantly better than others for that types. Which hand washing icon for example was the most effective using statistical tests for each dimension.

8. The authors collected some good data using rigorous data collection techniques but there is no rigorous analysis of the data as a reader would expect.

9. For aesthetics the authors state: “In terms of mean values, all samples scored between 5 and 6 (out of a total of 7) for the subjective design features, the distribution of scores over the mean did not differ much…” But without statistical tests, how do we know this is true?

10. In the correlation analysis it is unclear as to what is being correlated. This needs some major revision.

11. Ranks need some special statical procedures to deal with ordinal data.

ICONS

The authors did an excellent job of 120 Icons for the study that represent a number of important Covid prevention behaviors. Similarly, it is impressive that Icons were obtained from 26 countries potentially providing some cross cultural validity.

THEORETICAL BASE

The article need a broader theoretical base from the study of human communication to understand the basis of iconic communication. See a number of books on nonverbal communication for the conceptual and neurophysiological basis for iconic communication. The authors should provide at least a brief summary of literature on the value of icons from the literature of nonverbal communication. Their history and conceptual basis goes way beyond computer screens-they were used and studies in many contexts (e.g. traffic signs, medicine bottles, maps, product branding etc.) long before the advent of computer screens.

SAMPLE

Questionnaires were distributed sing the Star App buy we do not know if the participants were Chinese, European, American, global or what. This need clarification.

The authors provide a good table on characteristics of participants and good age distribution. However, we still do not know where they are from? Were these all from on particular country or was it an international sample? Was it a general sample of the Questionnaire Star App, or were some parameters specified. More detail is required.

Reviewer #2: Congratulations to the authors for their work on this study. The paper offers valuable insights into icon evaluation in public health. However, I have several comments and recommendations:

Introduction:

- Deepen the exploration of how iconography impacts public health communication. Highlight the influence of icon design on public behavior and decision-making, particularly during health crises.

- Incorporate a concise review of previous studies on public health iconography, emphasizing the novel contributions or challenges your research presents.

- Clearly articulate the theoretical framework guiding your study, linking it directly to your objectives and anticipated results.

Section 2.3 Evaluation Metrics for Icon Design:

- The categorization in Table 3 is somewhat unclear, particularly regarding the second and third categories. I recommend revising the descriptions to ensure they align with the text and avoid repetition.

Methodology:

- Clarify the criteria for selecting icons and the reasoning behind these choices to enhance the study's replicability.

- Provide a more detailed account of participant demographics to strengthen the study's validity.

- Detail the validation or pilot phase of the Questionnaire Star App, including languages used, and add relevant references.

- Explain the rationale for choosing the Bentler and Chou method.

- Clarify the process and purpose of the ranking test. Table 2 appears to present results rather than methodology, which could be confusing.

- Elaborate on the use of factor analysis and PCA, as they were mentioned but not detailed in the paper.

- For Table 2's categorization, consider supporting your approach with robust strategies like cluster analysis, Multidimensional Scaling (MDS), Content Analysis, or advanced AI techniques like CNNs, Autoencoders, Transfer Learning, or Deep Learning with Data Augmentation.

- Overall, a more thorough statistical analysis plan is needed, outlining the methods for data analysis and their alignment with your research questions.

6. PLOS authors have the option to publish the peer review history of their article (what does this mean?). If published, this will include your full peer review and any attached files.

Reviewer #1: **Yes: **Dr. Peter A Andersen

Reviewer #2: No

---

## [Author Response · Author response to Decision Letter 0]

23 May 2024

Reviewer #1: 

The study entitled: “The effects of icon design features on user perception and preference: A case study of icons for Covid-19” is an interesting study with some good stimulus material and good data collection. However, there are number of problems that would prevent publication of this study in Plos One. The major problem is a hodge podge of statistics that are in search of the question. The author could have and should have been able to determine which icons were most effective on their three dimensions within the group (e.g. for example for handwashing or mask wearing). That would have been useful information. So, I will begin with the major problems with the results and then discuss other problems.

RESULTS

The results need a major overhaul. The result lack statistical tests and most of the results are subjective and obtained through “eyeballing.”

1. The authors could have done a factor analysis and computed reliability coefficients for each of the 3 variables: communication effectiveness, design quality, and semantic distance but this is missing. The authors have psychometrically valid data for each of these dimensions but they do not use it. Since each of these scales contain multiple semantic differential items that represent the dimension an internal reliability coefficient is necessary and easy to compute. From that point some meaningful statistical tests could have been conducted and reported.

Reply: Thank you very much for your suggestion. We computed the reliability coefficients for the three variables as you suggested. Overall, the Cronbach’s α of Communication Effectiveness, Visual Design Perception, and Semantic Distance are 0.96, 0.987, 0.868, which, combined with the CITC values of the indicators, indicates that the data have a high quality of reliability, which can provide a reliable basis for further research and analysis. For details, please see P23-P24, Table 4.

2. We need to know if results for each dimension were aggregated and divided by the number of items that comprise the dimension or what. I cannot find scores for each dimension. This needs major revision and clarification.

Reply: Thank you very much for your constructive comments. We analyzed descriptive statistics for 12 indicators under three dimensions. Table 5 shows the characteristics of the data distribution for each indicator. The sample size is 223, and for each adjective variable, the minimum, maximum, mean, and standard deviation are given. For details, please see P24-P25, Table 5.

3. Just reporting the top 15 icon function types is subjective; were these significantly different?

Reply: I apologize, this may be a writing error on our part. The subjective rating test was performed on the 15 feature icons that were ranked first in the first stage (sorting test). The results of the ranking indicated the participants' preference for the icons, and participants were then asked to rate the icons on 12 subjective design features to further identify user-preferred icon design features.

4. What does it tell is that some function types are better than others? Does it mean that we should tell people to wash hands but not check their temperature? The research question needs to be posed and then systematically answered.

5. We need statistical evidence to back the authors’ assertion that: “it can be initially determined that users prefer icons that contain specific character images, actions.”

6. Similarly, the authors assert that “It is suggested that users prefer icons associated with images when real or illustrated more concretely.” What statistical evidence supports this assertion?

7. What would be interesting would be to ascertain for each type of icon which was significantly better than others for that types. Which hand washing icon for example was the most effective using statistical tests for each dimension.

Reply: Yes, your comments are crucial to our study, thank you very much. Based on your modifications 4, 5, 6, and 7, we have added the Friedman test. In order to compare users' preference for figurative images, we conducted a Friedman test so that we could interpret the data in a meaningful way. We classified icons into two categories: (1) figurative icons, i.e., icons containing specific characters, actions, and scenes, which are more concrete and vivid and have a sense of familiarity; and (2) abstract icons, i.e., generalized and simplified icons, which are more concise and abstract. In this study, three groups of icons with the same functions, which can represent the above two styles, were selected for testing again to examine whether there are significant differences between these two groups of icons on the 12 semantic scales. For details, please see P27-P30, Table 9.

8. The authors collected some good data using rigorous data collection techniques but there is no rigorous analysis of the data as a reader would expect. 

Reply: Thank you for pointing out the shortcomings, we have added as much data analysis as possible to support our research goals and questions. We hope that the revised manuscript will better reflect the value of the study.

9. For aesthetics the authors state: “In terms of mean values, all samples scored between 5 and 6 (out of a total of 7) for the subjective design features, the distribution of scores over the mean did not differ much…” But without statistical tests, how do we know this is true?

Reply: We apologize for the lack of rigor in our research. Regarding this, descriptive statistical analyses and one-sample t-tests were performed for these 12 semantic scales to compare their differences from the mean. For details, please see P24-P26, Table 5, Table 6 and Table 7.

10. In the correlation analysis it is unclear as to what is being correlated. This needs some major revision.

Reply: Thanks for pointing out this.Regarding this, we performed sperman correlation analyses for these 12 semantic scales to test the strength of the correlation between them. The results shows that there are significant inter-correlations between the 12 semantic scales. Among them, Recognizable and Concrete, Clear and Reliable, Reliable and Effective, Beautiful and Interesting, Interesting and Eye-Catching, Familiar and Eye-Catching, Eye-Catching and Organised had the highest correlation coefficients, i.e., 0.934, 0.933, 0.939, 0.941, 0.928, 0.939, 0.940 respectively. For details, please see P27,P29, Table 8.

11. Ranks need some special statical procedures to deal with ordinal data.

Reply: Thank you for your advice. We did as much data processing as we could, however, for the purposes of this study's objectives and questions, we focused our research on the second testing phase (subjective rating test).

ICONS

The authors did an excellent job of 120 Icons for the study that represent a number of important Covid prevention behaviors. Similarly, it is impressive that Icons were obtained from 26 countries potentially providing some cross cultural validity. 

Reply: Your valuable comments are greatly appreciated. Cross-cultural differences in icon understanding were not discussed in this study, which is a limitation of this study that we state in the conclusion section of the article.

THEORETICAL BASE

The article need a broader theoretical base from the study of human communication to understand the basis of iconic communication. See a number of books on nonverbal communication for the conceptual and neurophysiological basis for iconic communication. The authors should provide at least a brief summary of literature on the value of icons from the literature of nonverbal communication. Their history and conceptual basis goes way beyond computer screens-they were used and studies in many contexts (e.g. traffic signs, medicine bottles, maps, product branding etc.) long before the advent of computer screens.

Reply: Thanks to your comments, we have added relevant research literature to the first subsection of the literature review (iconic communication) and briefly stated the theoretical basis of iconic communication from the perspective of semiotics and cognitive psychology, as detailed on page 5 - 8.

SAMPLE

Questionnaires were distributed sing the Star App buy we do not know if the participants were Chinese, European, American, global or what. This need clarification.

The authors provide a good table on characteristics of participants and good age distribution. However, we still do not know where they are from? Were these all from on particular country or was it an international sample? Was it a general sample of the Questionnaire Star App, or were some parameters specified. More detail is required. 

Reply: Thank you for pointing out the details, which we have corrected.This study used random sampling method to select the study sample. A total of 300 participants aged between 18 and 40 years old, all of whom were residents of different parts of mainland China, were recruited to participate in the study. For more details, please see P13-14.

Reviewer #2: 

Congratulations to the authors for their work on this study. The paper offers valuable insights into icon evaluation in public health. However, I have several comments and recommendations:

Introduction:

- Deepen the exploration of how iconography impacts public health communication. Highlight the influence of icon design on public behavior and decision-making, particularly during health crises.

Reply: Thank you very much for pointing out the problem. We are taking your suggestions further to explore how iconography can influence public health communication, especially during health crises. Please check p2-p3 for more details.

- Incorporate a concise review of previous studies on public health iconography, emphasizing the novel contributions or challenges your research presents.

Reply: Thank you for your constructive comments, we have added research literature related to public health iconography in the second half of the introduction and emphasized the contributions and challenges of this study, please check p3-p5 for more details.

- Clearly articulate the theoretical framework guiding your study, linking it directly to your objectives and anticipated results.

Reply: Thanks to your comments, we have added relevant research literature to the first subsection of the literature review (iconic communication) and briefly stated the theoretical basis of iconic communication from the perspective of semiotics and cognitive psychology, as detailed on page 5- 8.

Section 2.3 Evaluation Metrics for Icon Design:

- The categorization in Table 3 is somewhat unclear, particularly regarding the second and third categories. I recommend revising the descriptions to ensure they align with the text and avoid repetition.

Reply: Thank you for your suggestion. The description of "design quality" is more general and seems to encompass all three, so we have changed the original "Design Quality" to "Visual Design Perception" for clarity. Please check page19 and Table 3 for more details.

Methodology:

- Clarify the criteria for selecting icons and the reasoning behind these choices to enhance the study's replicability.

Reply: Thank you very much for pointing out the issue.In order to be more representative and replicable, icons representing different design types, Image-related, Concept-related, Arbitrary, Semi-abstract, and Combined, were purposely selected for this study, excluding single textual and abbreviated types. In addition, textual descriptions were removed from the Combined icons when designing the questionnaire. Please check page15 for more details.

- Provide a more detailed account of participant demographics to strengthen the study's validity.

Reply: Thanks for your suggestion, we've added a detailed description of our demographic data as below:

“Table 1 shows the demographic characteristics of the participants. A total of 300 participants were recruited to participate in the study, all of whom were residents of different parts of mainland China. There was a balanced proportion of men and women among the participants, with men (n=102) accounting for 45.7% and women (n=121) accounting for 54.3%; the participants were all over 18 years old, and most of the participants were concentrated in the younger age group of 18-45 years old (n=152), which accounted for 68.2% of the participants; the participants' education level was concentrated in college, undergraduate and graduate students (n=185), which accounted for 82.9%, the good educational background indicates that the participant group has a good cognitive ability of icon design.”

Please see manuscript P14-15 and Table 1 for details.

- Detail the validation or pilot phase of the Questionnaire Star App, including languages used, and add relevant references.

Reply: Thank you for your constructive comments. It is very helpful to us. We have corrected the issues you raised in this subsection on Participants. Please see page 13.

- Explain the rationale for choosing the Bentler and Chou method.

Reply: Thank you for your comments. The rationale for the selection of the Bentler and Chou methods was explained as follows:

“Since the population of the parent group could not be determined, Bentler and Chou's method of determining sample size was used in this study. According to Bentler and Chou , the sample size is 5-10 times the number of questionnaire items when the parent group of the population is unknown, and the sample size can be enlarged by 20% by considering the number of invalid samples[42].”

Please see manuscript P13-14 for details.

- Clarify the process and purpose of the ranking test. Table 2 appears to present results rather than methodology, which could be confusing.

Reply: Yes, in order not to duplicate the presentation of all selected icons, we use a table (Table 2) to exhibit the 120 icons selected for the study and the results of the sorting test.

- Elaborate on the use of factor analysis and PCA, as they were mentioned but not detailed in the paper.

Reply:Thank you very much for pointing out the problem, we have explained in detail in subsection Statistical analysis the reasons for using factor analysis and principal component analysis in this study. Please check the subsection Statistical analysis, page 21-22.

- For Table 2's categorization, consider supporting your approach with robust strategies like cluster analysis, Multidimensional Scaling (MDS), Content Analysis, or advanced AI techniques like CNNs, Autoencoders, Transfer Learning, or Deep Learning with Data Augmentation.

Reply: Thank you for your suggestion. However, given the time and length constraints, as well as in terms of the objectives and questions of this study, we focused on the second testing phase (subjective scoring test).

- Overall, a more thorough statistical analysis plan is needed, outlining the methods for data analysis and their alignment with your research questions.

Reply: Thank you for pointing out the shortcomings, we have added as much data analysis as possible to support our research goals and questions. Please see manuscript P23-30 for more details.

We hope that the revised manuscript will better reflect the value of the study.

---

## [Editor Report · Decision Letter 1]

28 May 2024

The effects of icon design features on user perception and preference: A case study of icons for Covid-19

PONE-D-23-31275R1

Dear Dr. Lin,

We’re pleased to inform you that your manuscript has been judged scientifically suitable for publication and will be formally accepted for publication once it meets all outstanding technical requirements.

Kind regards,

Jorge Abelardo Falcón‐Lezama, PhD

Guest Editor

PLOS ONE
---

## [Editor Report · Acceptance letter]

24 Jun 2024

PONE-D-23-31275R1 

PLOS ONE

Dear Dr. Lin, 

I'm pleased to inform you that your manuscript has been deemed suitable for publication in PLOS ONE. Congratulations! Your manuscript is now being handed over to our production team.

Kind regards, 

on behalf of

Dr. Jorge Abelardo Falcón‐Lezama 

Guest Editor

PLOS ONE